# Health-Related Quality of Life among Cancer Survivors Depending on the Occupational Status

**DOI:** 10.3390/ijerph18073803

**Published:** 2021-04-06

**Authors:** Kisook Kim, Hyohyeon Yoon

**Affiliations:** Department of Nursing, Chung-Ang University, Seoul 06974, Korea; kiskim@cau.ac.kr

**Keywords:** health-related quality of life, cancer survivors, return to work, occupational status

## Abstract

The study aimed to identify and compare the factors affecting health-related quality of life (HRQoL) depending on the occupational status of cancer survivors. This study was a secondary data analysis from the Korea National Health and Nutrition Examination Survey (KNHANES) from 2014 to 2018. Hierarchical multivariate linear regression was used to investigate the factors affecting the HRQoL of each group. Non-working cancer survivors had significantly lower HRQoL than working cancer survivors (*p* < 0.001). A hierarchical multiple regression model showed that demographic, health-related, and psychological characteristics explained 62.0% of non-working cancer survivors’ HRQoL (F = 4.29, *p* < 0.001). Among the input variables, health-related characteristics were the most influential factors (ΔR^2^ = 0.274, F = 9.84, *p* < 0.001). For working cancer survivors, health-related characteristics were the only variable that was statistically associated with HRQoL (F = 5.556, *p* < 0.001). It is important to enhance physical activities and manage the chronic disease to improve the HRQoL of working cancer survivors. Further, managing health-related characteristics, including depressive symptoms and suicidal ideation, is necessary for non-working cancer survivors. Regarding working survivors, psychological factors such as depressive symptoms and suicidal tendencies did not affect HRQoL. Therefore, an early and effective return to work program should be developed for the improvement of their HRQoL.

## 1. Introduction

Due to the recent increase in early cancer detection and advanced cancer treatment technology, more than 16.9 million Americans (5% of the United States population) with a history of cancer were alive in 2019 [1]. The number of cancer survivors continues to increase despite an overall decline in age-standardized incidence rate [2]. With the increasing survival rate of cancer patients, there is more interest in the quality of life (QoL) of survivors after complete recovery along with the treatment [3].

Cancer survivors experience mental, physical, and economic difficulties and social role confusion from diagnosis to treatment [4]. Moreover, they experience physical and cognitive impairment, anxiety, and fear of cancer recurrence even after complete recovery [5]. Furthermore, physical and mental disorders can make it difficult to return to work, which can lead to economic difficulties [6]. It has been reported that these factors can negatively affect their QoL [7].

However, in prior studies, 40 to 70 percent of people who experienced traumatic events benefited from them [8], and some cancer survivors revealed that they were more grateful for their lives and their ability to overcome adversities improved after their journey from diagnosis to treatment [9]. Posttraumatic growth in cancer survivors has been shown a positive impact in improving QoL [10].

Cancer survivors are motivated to continue working or return to work after complete cancer treatment [11]. Returning to work is the first gateway for cancer survivors to improve their QoL and it means more than a source of income [12].

Return to work (RTW) is important for cancer survivors as it significantly affects their QoL [13] by harmonizing work and life, making them feel better [14], providing a sense of belonging and reintegration [15]. Moreover, not being able to work might interfere with daily life routines and lead to a lack of confidence and social isolation [14]. In a prior study, the QoL results showed significant differences between workers and non-workers with the latter having significantly reduced QoL [14]. The increasing value of work, perceived workability, job self-efficacy, and limiting fatigue are promising goals for promoting RTW [16]. Among them, workability and the effectiveness of the job are the key variables for predicting it.

When cancer is diagnosed at working age (between 16 and 64 years), survivors are faced with disabilities that can impair their ability to continue working [17]. Cognitive limitations and fatigue caused by cancer and various side effects of treatment procedures can limit physical ability, emotional health and cause functional disorders [18]. Therefore, various rehabilitation services and information are provided through professional, medical, physical, psychological, and educational interventions [19,20]. After completion of a multidisciplinary rehabilitation program, the RTW rate of cancer patients was high and they reported reduced fatigue and increased workability and QoL [21].

Successful RTW after cancer can be affected by employer-related factors, social support from coworkers and subjectively perceived work environment factors [22,23,24].

As the survival period of cancer survivors increases, their physical and psychological symptoms affect the QoL. Since cancer survivors have lower QoL compared to healthy people and other chronic patients [25], continued research on their QoL along with their health problems is important.

Furthermore, research on the factors associated with HRQoL depending on the occupational status reflecting various characteristics of cancer survivors can provide a more specific basis for improving the HRQoL of employed cancer survivors. This is the foundation for future long-term cancer patients’ RTW intervention strategies.

However, most studies on risk factors of cancer patients and the efficiency of RTW programs mainly focus on certain nations and continents such as Europe. Therefore, this study aimed to identify health-related factors that affect the HRQoL based on large-scale national research depending on the occupational status of cancer survivors.

## 2. Materials and Methods

### 2.1. Design

This study has a descriptive and correlational design to identify factors affecting the HRQoL depending on the occupational status of cancer survivors and to analyze factors that affect each type. This is a secondary data analysis study using data from the Korea National Health and Nutrition Examination Survey (KNHANES) for five years, 2014–2018 (https://knhanes.cdc.go.kr/knhanes/sub03/sub03_02_05.do (accessed on 1 April 2021)).

### 2.2. Participants

Raw data for five years from the sixth National Health and Nutrition Survey second year (2014) to the seventh National Health and Nutrition Survey third year (2018) were extracted according to the purpose of the study. These data had a composite sample design that stratified residents of 960 survey districts across the country by cities and towns, villages, and housing types, gender, age, residential area, head of household, and educational background; 36,931 people participated in the survey.

Among these subjects, 621 people revealed that they were diagnosed with gastric, liver, colon, breast, cervical, lung, thyroid, and other cancers at the time of the study; they were excluded. Based on the state of economic activity, the participants were classified into different groups: employed, unemployed, and inactive population. For example, if a person had gastric cancer and the answer for economic activity is yes (employed), he/she was selected as a study subject and classified into a group “with a job.”

Data of 618 subjects were used, excluding three cases in which the value of the HRQoL variable, the dependent variable of this study, was missing. Accordingly, 373 people in the group with jobs and 245 people in the group without jobs were selected as subjects for this study.

### 2.3. Measures

#### 2.3.1. Health-Related Quality of Life

For the HRQoL, raw data from the EuroQoL-5D (EQ-5D) survey approved by the EuroQoL Group (www.euroqol.org (accessed on 1 April 2021)) were used. On a three-point scale, for five items, “Mobility,” “Self-care,” “Usual activity,” “Pain/discomfort,” and “Anxiety/depression,” participants had to select “no problems,” “some problems,” or “extreme problems,” and measure them from 1 to 3 points. The HRQoL score was calculated by adding up the individual scores of each item, and the final score t was calculated by applying the following model. The closer the value is to 1, the better the HRQoL.
EQ5D = 1 − (0.05 + 0.096 × M2 + 0.418 × M3 + 0.046 × SC2 + 0.136 × SC3 + 0.051 × UA2 + 0.208 × UA3 + 0.037 × PD2 + 0.151 × PD3 + 0.043 × AD2 + 0.158 × AD3 + 0.05 × N3)

#### 2.3.2. Demographic and Health-Related Characteristics

For demographic and sociological factors, gender, age, education level, income level, average monthly gross household income, number of household members, and marital status were included. Current health-related factors were included for subjective health status, hypertension, osteoarthritis, osteoporosis, diabetes, and metabolic syndrome. We included activity restriction, presence of a disease in the previous one month, hospitalization in the previous one year, outpatient treatment and discomfort in the previous two weeks.

#### 2.3.3. Health Behaviors and Psychological Factors

Health behavior factors included unmet annual hospital service, preventive health behavior (flu vaccination, medical checkup), subjective body type recognition, drinking, smoking, walking, strength training, energy intake, intake of water, dietary fiber, sodium, and vitamin C as well as daily sleep time on weekends and weekdays were included. Psychological factors included current depressive symptoms, stress perception, depressive symptoms for two weeks or more, suicidal ideation/planning.

### 2.4. Ethical Considerations

The use of the original data from the KNHANES in this study adheres to the personal information protection and statistics law, and it provides the only data that cannot be estimated from the survey data. The researcher applied for the required information on the KCDC (Korea Centers for Disease Control and Prevention) website before starting the study. Moreover, the researcher downloaded the raw data after receiving approval to use the materials (https://knhanes.cdc.go.kr/ (accessed on 11 March 2021)).

### 2.5. Statistical Analysis

The data were analyzed using SPSS Statistics for Windows version 25.0 (IBM Corp., Armonk, NY, USA). The demographic characteristics and main variables of the participants were analyzed using descriptive statistics. Demographic, health-related characteristics, health behaviors and psychological characteristics were examined using *t*-tests and χ^2^ tests. Hierarchical multivariate linear regression was used to investigate the factors that affect HRQoL of each group.

## 3. Results

### 3.1. Comparison of Demographic and Health-Related Characteristics Depending on the Occupational Status of Cancer Survivors

In total, 615 cancer survivors were included in this study. Among them, 245 people were employed. Comparing the demographic characteristics of cancer survivors depending on their occupational status, there were statistically significant differences in the HRQoL score (*p* < 0.001), gender (*p* < 0.001), age (*p* < 0.001), educational level (*p* = 0.004), income level (*p* < 0.001), monthly household income (*p* < 0.001), and marital status (*p* < 0.001) (Table 1).

Regarding the HRQoL depending on the occupational status, the mean score of workers was 0.954, which was higher than that of the unemployed (0.877). The mean age of working cancer survivors was 56.56 years, which was relatively lower than that of non-working cancer survivors (64.28 years). The proportion of male workers was 50.6% which was higher than among non-workers (34.6%). The percentage of workers whose educational level was college or higher was 30.6% which was higher than that of non-workers (22.8%). Regarding the income level, 31.8% of workers belonged to the high-income category whereas among non-workers, it was only 19%. Further, workers had relatively higher household incomes than non-workers. When compared with non-working cancer survivors, workers were more likely to be married (88.2% for workers vs. 74.3% for non-workers).

Regarding health characteristics, there were statistically significant differences in subjective health status, prevalence of hypertension, osteoarthritis, osteoporosis, diabetes, activity restriction, prevalence of disease in the previous one month, outpatient treatment, and discomfort in the previous two weeks. When comparing subjective health status, more non-workers belonged to “very poor” than workers (19% for non-workers vs. 6.1% for workers). The prevalence of hypertension, osteoarthritis, osteoporosis, diabetes was statistically higher in non-workers.

### 3.2. Comparison of Health Behavior Characteristics Depending on the Occupational Status of Cancer Survivors

Regarding health-related behavior of workers and non-workers, there were statistically significant differences in unmet annual hospital service (*p* = 0.024), preventive health behavior (flu vaccination (*p* < 0.001), medical checkup (*p* = 0.004)), subjective body type recognition (*p* = 0.019), drinking (*p* < 0.001), smoking (*p* = 0.014), and energy intake (*p* = 0.018) (Table 2).

Higher percentages of non-workers than of workers exhibited “met annual hospital service” (10.5% for non-workers vs. 5.3% for workers) and “flu vaccination” (67.3% for non-workers vs. 49.8% for workers). Regarding subjective body recognition, “very thin” and “a little thin” showed higher percentages for non-workers than for workers. However, the percentage of “a little obese” was higher in workers than in non-workers. Further, we found that workers were more likely to smoke and drink than non-workers.

### 3.3. Comparison of Psychological Characteristics Depending on the Occupational Status of Cancer Survivors

As a result of psychological characteristics, there were statistically significant differences in current stress perception (*p* = 0.001), depressive symptoms present for two weeks or more (*p* = 0.010), suicidal ideation in the previous year (*p* = 0.006), and weekend sleep time (*p* = 0.019) (Table 3).

Regarding stress perception, the disparities between workers and non-workers were especially notable for “extremely high” (3.7% for worker vs. 7% for non-workers), “rather low” (62% for workers vs. 50.4% for non-workers), “almost none” (15.5% for workers vs. 27.1% for non-workers). In depressive symptoms present for two weeks or more, there was a higher percentage of non-workers than of workers (10.2% for non-workers vs. 4.5% for workers).

### 3.4. Multiple Regression Results with HRQoL as Dependent Variables

To identify the factors that affect HRQoL of cancer survivors, we performed hierarchical regression. To compare explanatory power and identify the influence of each variable, those with significant adjustments were entered as an independent variable in the regression model. We entered demographic characteristics in model 1, health-related characteristics in model 2, health behaviors in model 3, and psychological characteristics in model 4.

We performed hierarchical regression with working cancer survivors (Table 4). To test the assumption of linear regression, we examined the normality and linearity of all variables and the Durbin–Watson statistic was 0.934, thus eliminating the autocorrelation problem. Regression model 1 comprising working cancer survivors’ demographic characteristics explained approximately 11.7% of the variable, but it was not statistically significant (F = 1.972, *p* = 0.056). Health-related characteristics were additionally entered in model 2, which was significant (F = 5.556, *p* < 0.001), and the percentage of variance explained increased to 39.3%. Among them, hypertension (β = 0.183, *p* = 0.033) and angina pectoris (β = −0.204, *p* = 0.034) were significantly associated. Health behavior was entered in model 3. The percentage of variance explained increased to 47.5%, but it was not statistically significant (F = 1.409, *p* = 0.181). Finally, in model 4, psychological characteristics were entered. However, the model was also not statistically significant (F = 0.180, *p* = 0.982).

We performed hierarchical regression with non-working cancer survivors (Table 5). To test the assumption of linear regression, we examined the normality and linearity of all variables and the Durbin–Watson statistic was 1.341, thus eliminating the autocorrelation problem. Regression model 1, containing non-working cancer survivors’ demographic characteristics, was statistically significant (F = 6.671, *p* < 0.001) and explained approximately 24.4% of the variance. Among the demographic characteristics, marital status was statistically significant (β = 0.309, *p* = 0.004), and married people had a better HRQoL.

Health-related characteristics were entered in model 2, which was significant (F = 9.841, *p* < 0.001), and the percentage of variance explained increased to 51.8%, which is 27.4% higher than for model 1. Among the variables, angina pectoris (β = −0.154, *p* = 0.014), activity restriction status (β = −0.264, *p* < 0.001), discomfort in the previous two weeks (β = −0.232, *p* = 0.001) was related with HRQoL. People with angina pectoris, activity restriction, discomfort in the previous two weeks exhibited lower HRQoL. Health behavior was entered in model 3, and the percentage of variance explained increased to 54.9%, but it was not significant (F = 0.918, *p* = 0.525). Finally, in model 4, psychological characteristics were entered, and the model was significant (F = 4.298, *p* < 0.001). The final percentage of variance explained increased to 62% which was 7.1% higher than for model 3. The higher current depressive symptoms (β = −0.172, *p* = 0.011), suicidal ideation in the previous year (β = −0.193, *p* = 0.019) exhibited lower HRQoL. Hence, non-working cancer survivors who are single, with metabolic syndrome, discomfort in the previous two weeks, depressive symptoms, suicidal ideation in the previous year, have lower HRQoL.

## 4. Discussion

This study was conducted after dividing cancer survivors into groups, workers and non-workers, for identifying and comparing factors that affect HRQoL. The average HRQoL score of workers was 0.954 which was higher than that of non-workers (0.877). Similar results were presented in the Isaksson’s study, the QoL score of workers was higher than that of non-workers [14]. This could be construed as a basis to support the study result that working could positively affect the QoL of cancer survivors [13].

Our findings showed that male cancer survivors were more likely to have a job than female survivors. This result was in line with a previous study that concluded that married men return to work much faster than women [26]. It is because, while women tend to leave their job for the double burden of being responsible for family and work, men tend to continue working as they are the main source of income for their families [27].

The average age of workers was 56.56 years, which was lower than that of non-workers (64.28 years). This result was in line with findings of a previous study that showed young age is a predictor of RTW [28]. There was a significant difference in the employment status of cancer survivors in the level of education, income, and marital status. Low level of education and income led to low employment rate. Low education was related to physically demanding and low-paying work. Thus, cancer survivors who had physical difficulties due to cancer treatment were linked to a high chance of not working [29]. However, cancer survivors with a high level of education tend to readapt to the working conditions and easily coordinate, which affects their job availability [29]. The lower the level of education, the lower the income and percentage of employment. Consequently, it can lead to economic difficulties.

Our findings showed that non-workers assessed their subjective health status as “very poor” or “poor” significantly more often than workers. This result supported a previous study that those who answered “dissatisfaction” regarding their subjective health status had a worse workability assessment and physical and mental limitations. These findings were consistent with the result that subjective health status is also related to employment [30].

In the result of regression analysis, demographic characteristics were found as a significant predictor in the HRQoL of non-workers. Among them, marital status was a significant variable that affected the HRQoL of non-workers. It corresponded with the study result that spousal support plays a significant role in improving the QoL of aged cancer survivors [31]. Thus, it supports the study result that family environment and coping ability play a critical role in improving the overall QoL [32].

Health-related characteristics were significant predictors for HRQoL of workers and non-workers. As it was added in model 2, the percentage of variance explained increased to 46.6%, which was 27.4% higher than for model 1. It was the most influential variance. Among the health-related characteristics of workers, the presence of hypertension, angina pectoris, and osteoarthritis were significant variables in the HRQoL. This is consistent with the result of a study conducted on breast cancer survivors that showed chronic diseases, including diabetes, hypertension, and osteoarthritis occurring as side effects of cancer treatment could negatively affect the QoL [33]. Specifically, hypertension is more prevalent in cancer patients than in the general population. Cancer can directly affect the occurrence of hypertension or nephrotoxicity, a side effect of cancer therapy that can indirectly affect the occurrence of hypertension [34]. Similar results of a recent study showed that it lowers the QoL of cancer survivors [35].

Activity restriction was also found as a variable that influenced the HRQoL of workers and non-workers. Cancer survivors tend not to actively engage in many activities due to their psychological stress; this tendency leads to deterioration in cardiovascular health, muscle strength, and bone health which increases the risk of osteoporosis and cardiovascular disease. Eventually, cancer survivors experience restrictions in their activities and discomfort to daily life simultaneously [36]. Renal cancer survivors exhibited a much lower physical function than the general population [37]. As the physical function of cancer survivors has an influence on the QoL [38], it is believed that continuous physical management is necessary for the improvement of HRQoL.

The psychological characteristic was found to be a significant predictor of HRQoL of non-workers. Among them, depressive symptoms and suicidal ideation in the previous year are significant variables affecting the HRQoL. It supports the large-scale cohort studies’ result that negative psychosocial outcomes have a life-long impact on cancer survivors [39]. As cancer is a destructive experience in life, survivors complain about physical and physiological pain. It has also been found that the suicide rate of cancer survivors was higher than that of the general population [40]. A history of depression, mental illness [41], drug and chronic physical health are factors that lead to suicidal ideation [42]. It is necessary to conduct research in the multidisciplinary intervention to prevent psychological problems. The higher the depressive symptoms, the lower the QoL for cancer survivors. It is in line with the result of a study that showed cervical cancer survivors with depression have a low level of QoL [43].

Social support, family function, and physical health were associated with depression of cancer survivors [44]. Thus, non-workers were revealed to have higher levels of depression [44]. Consequently, non-workers tend to be socially isolated and struggle to find jobs and get social support making them vulnerable to depression [14]. Further, the result shows that non-workers have higher depressive symptoms for two weeks or more compared to workers. Based on prior research that cancer survivors experience depression [44] due to social role confusion and economic difficulties [4] during and after the treatment process [6], we can predict the absence of a job can cause depressive symptoms along with social isolation. Therefore, to improve HRQoL of cancer survivors, follow-up studies evaluating depressive symptoms and suicidal thoughts as high risk are necessary.

The results clearly show that workers have higher average QoL scores and lower stress and depressive symptoms than non-workers. Factors that affect the QoL of non-workers were marital status, activity restrictions, depressive symptoms, and suicidal ideation. The results show that QoL decreases when there is no spouse or family or social support and the person is physically unable to work. Non-worker cancer survivors often show poor outcomes in QoL, non-workers were related to impaired QoL, including depression [45], physical functioning, and role functioning, and non-workers with low income status have a worse QoL compared to workers [46]. Thus, it is necessary to consider strategies for promoting HRQoL that take into account various influence factors on non-workers. Furthermore, to improve the HRQoL of cancer survivors, it is necessary to secure social support and prepare for economic difficulty through continuous career training. Regarding working cancer survivors, psychological factors such as depressive symptoms and suicide planning did not affect the HRQoL. Therefore, an early and effective RTW program should be developed for the improvement of cancer survivors’ HRQoL.

### Strengths and Limitations

This study has certain limitations. First, since it was based on secondary data analysis, it could not reflect particular characteristics related to specific cancers. Second, our findings presented cross-sectional research which could not identify the causal relationship with RTW after cancer treatment.

Despite these limitations, this study has the advantage of conducting research based on large-scale national data, identifying factors that affect the HRQoL of cancer patients depending on the occupational status, and investigating causes. The study is significant in suggesting the need for a strategy for decent RTW for the improvement of HRQoL of cancer survivors.

## 5. Conclusions

The study aimed to identify the factors affecting HRQoL depending on the occupational status of cancer survivors based on the data of the Korea National Health and Nutrition Examination Survey (KNHANES). Health-related characteristics constitutes the only variable that significantly affects HRQoL of worker and non-workers. Demographic and psychological characteristics only affect the non-workers’ HRQoL. The findings show that promoting physical activity and managing the disease is essential to maintain workers’ health. Further, health-related characteristics and reducing depressive symptoms, and preventing suicide are necessary for non-workers. In this study, we identified overall predictive factors for the cancer survivors’ HRQoL. In the future research, it is necessary to distinguish characteristics by cancer types and identify factors that affect the HRQoL for cancer survivors.

## Figures and Tables

**Table 1 ijerph-18-03803-t001:** General characteristics and health-related variables depending on the occupational status, *n* = 618.

Variables	Categories	Working (*n* = 245)	Non-Working (*n* = 373)	χ^2^ or *t*	*p*
		*n* (%) or M ± SD			
HRQoL		0.954 ± 0.083	0.877 ± 0.171	−7.485	<0.001
Gender	Men	124 (50.6%)	129 (34.6%)	15.711	<0.001
	Women	121 (49.4%)	244 (65.4%)		
Age		56.56 ± 0.72	64.28 ± 11.06	8.220	<0.001
Education level	≤Elementary school	59 (24.1%)	139 (37.3%)	13.162	0.004
	Middle school	41 (16.7%)	49 (13.1%)		
	High school	70 (28.6%)	97 (26.0%)		
	≥College	75 (30.6%)	85 (22.8%)		
Income level	Low	33 (13.5%)	127 (34.0%)	37.790	<0.001
	Medium-low	64 (26.1%)	97 (26.0%)		
	Medium-high	69 (28.2%)	77 (20.6%)		
	High	78 (31.8%)	71 (19.0%)		
Monthly family income (approximate; in USD)		4171.44 ± 196.57	3042.37 ± 151.02	−4.610	<0.001
Number of household members	One	21 (8.6%)	56 (15.0%)	5.626	0.018
	Two or more	224 (91.4%)	317 (85.0%)		
Marital status	Married	216 (88.2%)	277 (74.3%)	20.516	<0.001
	Single and other	29 (11.8%)	96 (25.7%)		
Subjective health status	Excellent	8 (3.3%)	6 (1.6%)	28.399	<0.001
	Good	47 (19.2%)	45 (12.1%)		
	Fair	120 (49%)	150 (40.2%)		
	Poor	55 (22.4%)	101 (27.1%)		
	Very poor	15 (6.1%)	71 (19.0%)		
Hypertension	Yes	59 (24.1%)	139 (37.3%)	11.804	0.001
	No	186 (75.9%)	234 (62.7%)		
Osteoarthritis	Yes	27 (11.0%)	79 (21.2%)	10.740	0.001
	No	218 (89.0%)	294 (78.8%)		
Osteoporosis	Yes	17 (6.9%)	48 (12.9%)	5.525	0.019
	No	228 (93.1%)	325 (87.1%)		
Diabetes	Yes	27 (11.0%)	68 (18.2%)	5.909	0.015
	No	218 (89.0%)	305 (81.8%)		
Metabolic syndrome	Yes	67 (27.3%)	110 (29.5%)	0.332	0.564
	No	178 (72.7%)	263 (70.5%)		
Activity restriction	Yes	22 (9.0%)	90 (24.1%)	22.870	<0.001
	No	223 (91.0%)	283 (75.9%)		
Disease in the previous one month	Yes	23 (9.4%)	61 (16.4%)	6.110	0.013
	No	222 (90.6%)	312 (83.6%)		
Hospitalization in the previous one year	Yes	72 (29.4%)	119 (31.9%)	0.438	0.508
	No	173 (70.6%)	254 (68.1%)		
Outpatient treatment in the previous two weeks	Yes	84 (34.3%)	174 (46.6%)	9.294	0.002
	No	161 (65.7%)	199 (53.4%)		
Discomfort in the previous two weeks	Yes	50 (20.4%)	127 (34.0%)	13.461	<0.001
	No	195 (79.6%)	246 (66.0%)		

HRQoL = health-related quality of life.

**Table 2 ijerph-18-03803-t002:** Health behavior-related variables depending on the occupational status, *n* = 618.

Variables	Categories	Working (*n* = 245)	Non-Working (*n* = 373)	χ^2^ or *t*	*p*
		*n* (%) or M ± SD			
Annual hospital service unmet	Unmet	232 (94.7%)	334 (89.5%)	5.089	0.024
	Met	13 (5.3%)	39 (10.5%)		
Flu vaccination	Yes	122 (49.8%)	251 (67.3%)	18.918	<0.001
	No	123 (50.2%)	122 (32.7%)		
Medical checkup	Yes	188 (76.7%)	246 (66.0%)	8.223	0.004
	No	57 (23.3%)	127 (34.0%)		
Subjective body type recognition	Very thin	12 (4.9%)	30 (8.0%)	11.765	0.019
	A little thin	23 (9.4%)	58 (15.5%)		
	Normal	105 (42.9%)	167 (44.8%)		
	A little obese	87 (35.5%)	99 (26.5%)		
	Very obese	18 (7.3%)	19 (5.1%)		
Drinking	Yes	224 (91.4%)	280 (75.1%)	26.313	<0.001
	Never	21 (8.6%)	93 (24.9%)		
Smoking	Yes	27 (11.0%)	21 (5.6%)	5.998	0.014
	No	218 (89.0%)	352 (94.4%)		
Walking in the previous week	Yes	194 (79.2%)	303 (81.2%)	0.395	0.530
	No	51 (20.8%)	70 (18.8%)		
Strength training in the previous week	Yes	64 (26.1%)	83 (22.3%)	1.222	0.269
	No	181 (73.9%)	290 (77.7%)		
Energy intake (Kcal)		89.93 ± 11.92	58.16 ± 6.04	−2.380	0.018
Water intake (g)		12.17 ± 2.74	10.65 ± 1.73	−0.470	0.639
Dietary fiber intake (g)		0.39 ± 0.06	0.54 ± 0.07	1.550	0.123
Sodium intake (mg)		81.69 ± 25.44	53.37 ± 11.96	−1.010	0.315
Vitamin C intake (mg)		0.26 ± 0.08	0.49 ± 0.2	1.060	0.292

**Table 3 ijerph-18-03803-t003:** Psychology-related variables depending on the occupational status, *n* = 618.

Variables	Categories	Working (*n* = 245)	Non-Working (*n* = 373)	χ^2^ or *t*	*p*
		*n* (%) or M ± SD			
Sleep time during weekends (min/day)		450.88 ± 6.63	455.71 ± 6.13	2.360	0.019
Sleep time during weekdays (min/day)		420.43 ± 5.69	439.96 ± 5.98	0.530	0.599
Current depressive symptoms	Yes	7 (2.9%)	19 (5.1%)	1.836	0.175
	No	238 (97.1%)	354 (94.9%)		
Stress perception	Extremely high	9 (3.7%)	26 (7.0%)	16.191	0.001
	High	46 (18.8%)	58 (15.5%)		
	Rather low	152 (62.0%)	188 (50.4%)		
	Almost none	38 (15.5%)	101 (27.1%)		
Depressive symptoms present for two weeks or more	Yes	11 (4.5%)	38 (10.2%)	6.576	0.010
	No	234 (95.5%)	335 (89.8%)		
Suicidal ideation in the previous year	Yes	2 (0.8%)	18 (4.8%)	7.591	0.006
	No	243 (99.2%)	355 (95.2%)		
Suicidal planning in the previous year	Yes	4 (1.6%)	10 (2.7%)	0.734	0.392
	No	241 (98.4%)	363 (97.3%)		

**Table 4 ijerph-18-03803-t004:** Factors influencing health-related quality of life of working cancer survivors, *n* = 245.

**Variables**	**Categories**	**Model 1**			**Model 2**			**Model 3**			**Model 4**		
		**b**	**β**	***t*(*p*) **	**b**	**β**	***t*(*p*) **	**b**	**β**	***t*(*p*) **	**b**	**β**	***t*(*p*) **
	(Contrast)	1.022		13.02 (<0.001)	1.076		14.99 (<0.001)	1.046		10.67 (<0.001)	1.043		8.96 (<0.001)
General characteristics	Age	−0.002	−0.217	−1.82 (0.071)	−0.001	−0.099	−0.86 (0.390)	−0.001	−0.109	−0.86 (0.392)	−0.001	−0.096	−0.70 (0.487)
	Gender	0.004	0.024	0.25 (0.800)	−0.009	−0.056	−0.63 (0.531)	−0.018	−0.105	−1.10 (0.274)	−0.019	−0.115	−1.14 (0.257)
	Educational level	−0.021	−0.296	−2.29 (0.024)	−0.016	−0.218	−1.89 (0.062)	−0.017	−0.234	−1.92 (0.058)	−0.015	−0.216	−1.66 (0.101)
	Income quintile	0.013	0.162	1.00 (0.322)	0.002	0.028	0.18 (0.855)	−0.003	−0.041	−0.25 (0.799)	−0.005	−0.058	−0.34 (0.735)
	Monthly family income	0.000	0.182	1.08 (0.283)	0.000	0.144	0.95 (0.346)	0.000	0.200	1.22 (0.225)	0.000	0.188	1.09 (0.279)
	Home ownership status	0.001	0.007	0.07 (0.943)	−0.002	−0.016	−0.19 (0.849)	−0.004	−0.030	−0.35 (0.726)	−0.003	−0.026	−0.29 (0.771)
	Number of household members	−0.001	−0.004	−0.03 (0.973)	−0.011	−0.043	−0.43 (0.666)	−0.021	−0.085	−0.83 (0.410)	−0.032	−0.129	−1.08 (0.283)
	Marital status	0.017	0.070	0.63 (0.531)	0.034	0.141	1.39 (0.166)	0.048	0.195	1.80 (0.076)	0.055	0.226	1.87 (0.065)
Health-related characteristics	Subjective health status				−0.015	−0.156	−1.84 (0.069)	−0.012	−0.123	−1.38 (0.169)	−0.011	−0.115	−1.20 (0.232)
	Hypertension				0.034	0.183	2.15 (0.033)	0.033	0.176	2.08 (0.040)	0.031	0.168	1.86 (0.066)
	Angina pectoris				−0.050	−0.204	−2.15 (0.034)	−0.031	−0.129	−1.25 (0.216)	−0.034	−0.140	−1.28 (0.202)
	Osteoarthritis				−0.027	−0.104	−1.20 (0.232)	−0.033	−0.129	−1.30 (0.195)	−0.033	−0.130	−1.25 (0.216)
	Metabolic syndrome				−0.003	−0.017	−0.21 (0.831)	0.003	0.017	0.19 (0.854)	0.004	0.025	0.26 (0.798)
	Activity restriction status				−0.054	−0.204	−2.28 (0.024)	−0.036	−0.137	−1.44 (0.153)	−0.038	−0.144	−1.41 (0.162)
	General characteristics				−0.019	−0.070	−0.88 (0.383)	−0.032	−0.121	−1.47 (0.144)	−0.025	−0.093	−1.04 (0.302)
	Outpatient treatment in the previous two weeks				−0.012	−0.066	−0.77 (0.441)	−0.013	−0.073	−0.84 (0.403)	−0.012	−0.067	−0.73 (0.465)
	Discomfort in the previous two weeks				−0.039	−0.195	−2.08 (0.040)	−0.041	−0.203	−2.06 (0.042)	−0.042	−0.212	−2.03 (0.045)
Health behaviors	Unmet annual hospital service							−0.017	−0.040	−0.49 (0.623)	−0.018	−0.043	−0.50 (0.621)
	Flu vaccination							0.004	0.022	0.24 (0.814)	0.001	0.007	0.07 (0.942)
	Medical checkup							−0.008	−0.037	−0.44 (0.661)	−0.011	−0.052	−0.60 (0.551)
	Subjective body type recognition							−0.006	−0.064	−0.70 (0.488)	−0.005	−0.054	−0.56 (0.578)
	Weight control in the previous year							0.003	0.048	0.54 (0.594)	0.003	0.047	0.49 (0.624)
	Drinking							0.031	0.112	1.17 (0.243)	0.030	0.109	1.09 (0.279)
	Daily high-intensity physical activity							0.002	0.002	0.03 (0.975)	0.000	0.000	0.01 (0.995)
	Walking in the previous week							0.054	0.266	3.06 (0.003)	0.055	0.271	3.00 (0.003)
	Muscular exercise in the previous week							0.014	0.078	0.90 (0.373)	0.016	0.089	0.97 (0.333)
	Understanding of the nutritional labeling of food							−0.005	−0.025	−0.25 (0.800)	−0.004	−0.023	−0.21 (0.837)
	Vitamin C intake							−0.007	−0.082	−1.04 (0.300)	−0.007	−0.090	−1.08 (0.284)
Psychological characteristics	Current depressive symptoms										0.003	0.007	0.08 (0.935)
	Perceived stress level										0.003	0.023	0.24 (0.810)
	Depression for more than two weeks										−0.040	−0.085	−0.85 (0.395)
	Suicidal ideation in the previous year										0.023	0.024	0.28 (0.782)
	Suicidal planning in the previous year										−0.006	−0.010	−0.11 (0.916)
	Sleep time during weekends										0.000	0.010	0.12 (0.904)
	R^2^	0.117			0.393			0.475			0.481		
	Adj. R^2^ (ΔR^2^)	0.058 (0.117)			0.299 (0.276)			0.327 (0.082)			0.292 (0.006)		
	F(*p*)	1.972 (0.056)			5.556 (<0.001)			1.409 (0.181)			0.180 (0.982)		

**Table 5 ijerph-18-03803-t005:** Factors influencing health-related quality of life of non-working cancer survivors, *n* = 373.

**Variables**	**Categories**	**Model 1**			**Model 2**			**Model 3**			**Model 4**		
		**b**	**β**	***t*(*p*) **	**b**	**β**	***t*(*p*) **	**b**	**β**	***t*(*p*) **	**b**	**β**	***t*(*p*) **
	(Contrast)	0.727		5.83 (<0.001)	0.916		7.64 (<0.001)	0.958		6.51 (<0.001)	0.902		5.45 (<0.001)
General characteristics	Age	−0.001	−0.037	−0.44 (0.660)	−0.001	−0.035	−0.46 (0.647)	0.000	−0.009	−0.10 (0.919)	0.000	0.012	0.13 (0.893)
	Gender	0.029	0.078	1.00 (0.318)	0.013	0.034	0.47 (0.636)	−0.009	−0.025	0.32 (0.751)	−0.011	−0.029	−0.38 (0.703)
	Educational level	−0.004	−0.026	−0.30 (0.761)	0.000	−0.003	−0.04 (0.965)	0.001	0.008	0.11 (0.915)	0.006	0.039	0.55 (0.585)
	Income quintile	0.039	0.244	1.62 (0.108)	0.019	0.121	0.96 (0.340)	0.007	0.045	0.34 (0.737)	0.002	0.012	0.10 (0.924)
	Monthly family income	0.000	0.045	0.31 (0.756)	0.000	−0.004	−0.03 (0.974)	0.000	0.026	0.21 (0.836)	0.000	0.025	0.21 (0.831)
	Home ownership status	0.005	0.018	0.24 (0.813)	0.026	0.090	1.41 (0.159)	0.021	0.073	−1.07 (0.288)	0.022	0.077	1.15 (0.251)
	Number of household members	−0.004	−0.008	−0.09 (0.931)	−0.033	−0.069	−0.85 (0.399)	−0.045	−0.095	−1.09 (0.279)	−0.049	−0.103	−1.22 (0.224)
	Marital status	0.127	0.309	2.96 (0.004)	0.119	0.290	3.33 (0.001)	0.134	0.326	3.50 (0.001)	0.136	0.332	3.69 (<0.001)
Health-related characteristics	Subjective health status				−0.019	−0.106	−1.59 (0.113)	−0.019	−0.105	−1.47 (0.142)	−0.008	−0.045	−0.64 (0.522)
	Hypertension				0.018	0.048	0.74 (0.458)	0.010	0.025	0.37 (0.712)	−0.018	−0.047	−0.70 (0.484)
	Angina pectoris				−0.069	−0.154	−2.48 (0.014)	−0.073	−0.163	−2.52 (0.013)	−0.072	−0.161	−2.53 (0.013)
	Osteoarthritis				0.008	0.015	0.25 (0.807)	0.008	0.014	0.22 (0.827)	0.003	0.006	0.09 (0.927)
	Metabolic syndrome				0.038	0.098	1.59 (0.122)	0.043	0.112	1.70 (0.091)	0.055	0.141	2.21 (0.029)
	Activity restriction status				−0.109	−0.264	−3.96 (<0.001)	−0.111	−0.268	−3.83 (<0.001)	−0.115	−0.279	−4.16 (<0.001)
	General characteristics				−0.058	−0.114	−1.75 (0.082)	−0.049	−0.096	−1.39 (0.165)	−0.025	−0.050	−0.70 (0.485)
	Outpatient treatment within the previous two weeks				0.002	0.006	0.09 (0.928)	−0.009	−0.024	−0.37 (0.712)	−0.011	−0.030	−0.48 (0.634)
	Discomfort in the previous two weeks				−0.089	−0.232	−3.25 (0.001)	−0.078	−0.201	−2.64 (0.009)	−0.061	−0.158	−2.14 (0.034)
Health behaviors	Unmet annual hospital service							−0.013	−0.022	−0.31 (0.754)	−0.005	−0.009	−0.14 (0.889)
	Flu vaccination							−0.010	−0.027	−0.42 (0.678)	−0.015	−0.041	−0.65 (0.518)
	Medical checkup							0.045	0.118	1.96 (0.051)	0.033	0.085	1.46 (0.148)
	Subjective body type recognition							−0.012	−0.068	−0.97 (0.336)	−0.009	−0.049	−0.72 (0.471)
	Weight control in the previous year							0.000	0.003	0.04 (0.965)	−0.005	−0.036	−0.56 (0.579)
	Drinking							0.004	0.011	0.17 (0.869)	0.005	0.012	0.20 (0.843)
	Daily high-intensity physical activity							−0.099	−0.042	−0.68 (0.500)	−0.114	−0.048	−0.81 (0.417)
	Walking in the previous week							−0.010	−0.022	−0.33 (0.741)	−0.006	−0.012	−0.18 (0.855)
	Muscular exercise in the previous week							0.036	0.081	1.28 (0.203)	0.045	0.102	1.69 (0.093)
	Understanding of the nutritional labeling of food							−0.011	−0.031	−0.40 (0.688)	−0.024	−0.064	−0.87 (0.388)
	Vitamin C intake							−0.005	−0.093	−1.46 (0.147)	−0.005	−0.096	−1.39 (0.166)
Psychological characteristics	Current depressive symptoms										−0.156	−0.172	−2.58 (0.011)
	Perceived stress level										0.007	0.031	0.45 (0.656)
	Depression for more than two weeks										−0.001	−0.001	−0.02 (0.987)
	Suicidal ideation in the previous year										−0.155	−0.193	−2.38 (0.019)
	Suicidal planning in the previous year										0.041	0.038	0.53 (0.600)
	Sleep time during weekends										0.000	0.012	0.19 (0.846)
	R^2^	0.244			0.518			0.549			0.620		
	Adj. R^2^ (ΔR^2^)	0.208 (0.244)			0.466 (0.274)			0.462 (0.031)			0.527 (0.071)		
	F(*p*)	6.671 (<0.001)			9.841 (<0.001)			0.918 (0.525)			4.298 (<0.001)		

## Data Availability

Not applicable.

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
