# Peer review of "Health-Related Quality of Life among Cancer Survivors Depending on the Occupational Status"

_ijerph, 2021, doi:10.3390/ijerph18073803_

Round 1
Reviewer 1 Report
The author presented an interesting statistical evaluation of Health-related quality of life for cancer survivors in relationship to their occupational status.
The paper is well organized and written, I have only some short comments.
1) Lines 99-100: the authors could avoid to repeat the word "cancer" for each organ.
2) Line 102: there is a red dot after "...inactive population".
3) Table 2 could be formatted like the others, by dividing each listed variables and corresponding data with horizontal lines.
4) Table 3: I think the formatting should be improved, the whole table is not completely visible.
5) I want to advise the correction of this sentence, lines 295-296: I've put in bold 2 words that I think should be part of the same word: "Eventually, cancer survivors experience res depressive symptoms presented trictions in their activities and discomfort to daily life simultaneously [36]."
Author Response
Response to Reviewer 1 Comments
Point 1: Lines 99-100: the authors could avoid to repeat the word "cancer" for each organ.
Response 1: Thank you for your detailed comment. I deleted the word ‘cancer’ for each organ as follows:
“gastric cancer, liver cancer, colon cancer, breast cancer, cervical cancer, lung cancer, thyroid cancer, and other cancers’ were corrected ‘gastric, liver, colon, breast, cervical, lung, thyroid, and other cancers.”
Point 2: Line 102: there is a red dot after "...inactive population".
Response 2: It was corrected to black.
Point 3: Table 2 could be formatted like the others, by dividing each listed variables and corresponding data with horizontal lines.
Response 3: Based on your opinion, I divided each listed variables and corresponding data with horizontal lines, and split table 2 into two tables to organize neatly.
Point 4: Table 3: I think the formatting should be improved, the whole table is not completely visible.
Response 4: As per your suggestion, we revised the table.
Point 5: I want to advise the correction of this sentence, lines 295-296: I've put in bold 2 words that I think should be part of the same word: "Eventually, cancer survivors experience res depressive symptoms presented trictions in their activities and discomfort to daily life simultaneously [36]."
Response 5: According to your comment, we corrected this sentence as follows: “Eventually, cancer survivors experience restrictions in their activities and discomfort to daily life simultaneously.”

Reviewer 2 Report
Comments
In this article, the authors report the Health-related QoL among cancer survivors based on the occupational status.
- Abstract: Written well
- Introduction: Overall written well but it's too long. The authors are suggested cutting down the length.
- Methods: Provide details about what constitutes employed (self or under an employer). Are factors such as socioeconomic status and out-of-pocket costs (PMID: 30191078) required for health care available? I see that income level, educational level, monthly income was reported. As these are directly related to working status, they are important to report. If not able to report them in the limitation section. Table 2: There is a multitude of variables that are either no much relevant or move ot the supplementary section. It's making the table very busy
- Furthermore, is insurance status a variable in this database? Multiple studies have reported the insurance in receipt (refer to PMID: 30191078) of cancer prevention and further care. This should be noted in the manuscript and discussion. Additionally, barriers for receipt of cancer care in nonworker group is important. Perhaps the authors can reports which contribute to poor outcomes in this non-workers group (low income, lack of a job, low self-perception of health, and overall poor outcomes). This helps the audience to perceive the importance of factoring in the factors from a public health point of view.
- Provide limitations for this manuscript. In the methods section: Provide links to this database and variables available.
Author Response
Response to Reviewer 2 Comments
Point 1: Introduction: Overall written well but it's too long. The authors are suggested cutting down the length.
Response 1: Thank you for your valuable comment. Some of the contents of the introduction have been reorganized and repeated meaning contents have been deleted.
Point 2: Methods: Provide details about what constitutes employed (self or under an employer).
Response 2: This study is a secondary data analysis study based on the data of the national study, and the inclusion of variables was limited.
The data we analysed does not included information on whether the survivor is self or under employer.
Point 3: Are factors such as socioeconomic status and out-of-pocket costs (PMID: 30191078) required for health care available? I see that income level, educational level, monthly income was reported. As these are directly related to working status, they are important to report. If not able to report them in the limitation section.
Response 3: Because of the Korean national insurance system, 97.2% of Koreans are covered by national health insurance. Due to the characteristics of the Korean health care system, socioeconomic status does not influence much on the health care availability.
I looked for research related to this, and there was one study showing the result that breast cancer survivor Participants with national health insurance coverage had a higher clinical preventive service use rate than those with occupational health insurance (Kim, 2020), but no other related to health care availability. However, since what you mentioned is also very important factors, I will put it as s variable in the next study.
Point 4: Table 2: There is a multitude of variables that are either no much relevant or move to the supplementary section. It's making the table very busy.
Response 4: As mentioned in the Measure section, variables in the table2 are identified relevants in the previous research. In this study, we found that there were no statistically significant differences in water, dietary fiber, sodium intake and sleep time during weekdays according to occupational status of cancer survivors. Per your comment, we split table 2 into two tables for easier viewing.
Point 5: Furthermore, is insurance status a variable in this database? Multiple studies have reported the insurance in receipt (refer to PMID: 30191078) of cancer prevention and further care. This should be noted in the manuscript and discussion.
Response 5: As mention in point 2 and 3, this data is secondary data analysis and the inclusion of variables was limited. Since 97.2% of Koreans are covered by national health insurance, insurance status does not influence much on the result of the research.
Point 6: Additionally, barriers for receipt of cancer care in nonworker group is important. Perhaps the authors can reports which contribute to poor outcomes in this non-workers group (low income, lack of a job, low self-perception of health, and overall poor outcomes). This helps the audience to perceive the importance of factoring in the factors from a public health point of view.
Response 6: According to your suggestion, we added this sentence as follows:
“Non-worker cancer survivors often show poor outcomes in QoL, non-workers were related to impaired QoL, including depression [45], physical functioning and role functioning and non-workers with low income status have a worse QoL compared to the workers [46]. Thus, it is necessary to consider strategies for promoting HRQoL that take into account various influence factors on non-workers.”
Point 7: Provide limitations for this manuscript. In the methods section: Provide links to this database and variables available.
Response 7: Based on your comment, we reorganized to stand out the limitation and added links to this database in design section.
Reference : Kim, K. Use of Clinical Preventive Service and Related Factors in Middle-Aged Postmenopausal Women in Korea. Healthcare (Basel). 2020, 8(2)
